# Postural Instability and Risk of Falls in Patients with Parkinson’s Disease Treated with Deep Brain Stimulation: A Stabilometric Platform Study

**DOI:** 10.3390/brainsci13091243

**Published:** 2023-08-25

**Authors:** Giorgio Leodori, Marco Santilli, Nicola Modugno, Michele D’Avino, Maria Ilenia De Bartolo, Andrea Fabbrini, Lorenzo Rocchi, Antonella Conte, Giovanni Fabbrini, Daniele Belvisi

**Affiliations:** 1IRCCS Neuromed, 86077 Pozzilli, Italy; giorgio.leodori@uniroma1.it (G.L.); fkt.santillimarco@gmail.com (M.S.); nicusmod@gmail.com (N.M.); davinomichele87@gmail.com (M.D.); antonella.conte@uniroma1.it (A.C.); daniele.belvisi@uniroma1.it (D.B.); 2Department of Human Neuroscience, University of Rome “Sapienza”, 00185 Rome, Italy; mariailenia.debartolo@uniroma1.it (M.I.D.B.); andrea.fabbrini@uniroma1.it (A.F.); 3Department of Medical Sciences and Public Health, University of Cagliari, 09042 Cagliari, Italy; l.rocchi@ucl.ac.uk

**Keywords:** deep brain stimulation, fall risk, falls, Parkinson’s disease, postural instability, stabilometric platform, movement disorders, postural control

## Abstract

Postural instability (PI) in Parkinson’s disease (PD) exposes patients to an increased risk of falls (RF). While dopaminergic therapy and deep brain stimulation (DBS) improve motor performance in advanced PD patients, their effects on PI and RF remain elusive. PI and RF were assessed using a stabilometric platform in six advanced PD patients. Patients were evaluated in OFF and ON dopaminergic medication and under four DBS settings: with DBS off, DBS bilateral, and unilateral DBS of the more- or less-affected side. Our findings indicate that dopaminergic medication by itself exacerbated PI and RF, and DBS alone led to a decline in RF. No combination of medication and DBS yielded a superior improvement in postural control compared to the baseline combination of OFF medication and the DBS-off condition. Yet, for ON medication, DBS significantly improved both PI and RF. Among DBS conditions, DBS bilateral provided the most favorable outcomes, improving PI and RF in the ON medication state and presenting the smallest setbacks in the OFF state. Conversely, the more-affected side DBS was less beneficial. These preliminary results could inform therapeutic strategies for advanced PD patients experiencing postural disorders.

## 1. Introduction

Postural control relies on the integrated processing of multisensory afferent information by the vestibular, proprioceptive, and visual system. These are integrated in the cerebellum and basal ganglia and result in output to the brainstem and spinal cord for the execution of postural reflexes aimed at maintaining balance [1]. These mechanisms are deranged in the advanced stages of Parkinson’s disease (PD), which are often characterized by postural instability (PI). Abnormal balance control in PD patients may manifest either during stance, where abnormal posture may be observed, or in dynamic conditions, characterized by poor automatic postural reactions and festination [2]. PI exposes PD patients to an increased risk of falls (RF), which often leads to a dramatic worsening of the patient’s clinical status [3]. In PD, PI recognizes a multifactorial origin that includes muscle stiffness, abnormal visual, vestibular, and proprioceptive processing, as well as cognitive deficits [4]. These abnormalities result in a mismatch between the center of mass (COM) and the center of pressure (COP), with a consequent increase in RF [5]. While the efficacy of deep brain stimulation (DBS) of the subthalamic nucleus (STN) in improving appendicular symptoms in PD is widely accepted, its effect on axial symptoms is still not clear [6]. Limited evidence has suggested a postural improvement after DBS [7,8], whereas other work reported either a worsening [9] or no effect [10,11]. Different methodologies might contribute to the observed variability in outcomesOnly a small number of studies have employed objective posturometric measurements to assess the effects of Deep Brain Stimulation (DBS) on postural control [6,12,13,14,15,16]. While these studies provide some insight, their findings are limited and inconsistent

We hypothesized that these inconsistencies could be due, at least in part, to the fact that possible interactions between DBS and dopaminergic medications were not fully considered. In line with our hypothesis, recent evidence suggested a potential synergistic effects between STN-DBS and dopaminergic treatment on postural control in PD [17]. Furthermore, while the bilateral effects of DBS in PD have been widely explored, the outcomes from unilateral DBS remain less understood [18]. Evidence suggested potential lateralized effects of STN DBS on axial motor symptoms [19]. The exploration of these unilateral effects can provide deeper insights into optimizing therapeutic strategies for PD patients. Finally, only a subset of these studies employed objective posturometric measurements, which might have contributed further to the inconclusive results observed across the literature.

Therefore, the aim of our study was to systematically evaluate the effects of different DBS settings, dopaminergic therapy, and their interaction on PI and RF parameters, assessed using a stabilometric platform. Understanding which DBS/medication combinations result in greater postural stability may lead to new strategies aimed at improving axial symptoms in PD.

## 2. Materials and Methods

### 2.1. Participants

We enrolled six consecutive patients with PD from the Parkinson’s disease center of the IRCCS Neuromed Institute (Pozzilli, Italy). Inclusion criteria included a diagnosis of PD based on international clinical criteria [20], disease duration longer than 6 years, bilateral STN DBS performed more than 1 year before the evaluation, and confirmed STN leads position. Exclusion criteria were a history of neurological conditions other than PD, significant disorders of visual acuity vestibular or proprioceptive dysfunction, major orthopedic conditions, diabetes mellitus, and Mini-Mental State Examination (MMSE) score < 24. The study was performed according to the Declaration of Helsinki. Patients signed an informed consent form prior to examination, and the study was approved by the local ethics committee.

### 2.2. Clinical Assessment

Clinical evaluation included collection of patients’ demographics, disease duration, Levodopa equivalent daily dose (LEDDs), Unified Parkinson’s disease rating scale (UPDRS) part III, MMSE. LEDD is a conversion factor that allows one to estimate the global amount of dopaminergic treatment by converting each antiparkinsonian drug dosage into levodopa equivalent. UPDRS part III provides an objective estimation of motor symptom severity in PD based on the evaluation of the examiner.

### 2.3. Stabilometric Platform

We assessed PI and RF using the Biodex Stability System (BSS) (Biodex, Inc., Shirley, NY, USA), a standardized system that allows for the objective and reliable quantification of postural control in static and dynamic conditions [21,22]. The BSS consists of a circular platform that is free to move in the anterior–posterior and medial–lateral axes with 12 levels of stability. The system provides the degree and velocity of platform tilt and, given the subject’s height, it can measure the COM displacement. Digitized COM and platform tilt are transmitted to dedicated software for objective quantification of PI and RF (Version 1.08, Biodex Inc., New York, NY, USA).

### 2.4. PI and RF Assessment

PI and RF were assessed using standardized tests provided by the system’s dedicated software [21,22]. At the beginning of the test session, the patient was asked to stand on the platform in a stable position. The feet position was marked to ensure a consistent starting position across conditions, thus excluding confounding due to a different base of support. PI was assessed by measuring patients’ COM sway during stance with a stable platform. To assess RF, patients were instructed to maintain a stable posture while the platform became progressively more unstable. RF was described as patients’ ability to control the platform angle and was quantified as degree variance over time. Participants performed three consecutive trials for each test and in each experimental condition. PI and RF were scored from 0 to 18, each score representing the mean of the three measurements. Higher scores corresponded to poorer performance. One training trial for each test was performed at the beginning of the experimental session and its value discarded. Two examiners were ready to support the patient in the event of a fall; in this case, a score of 18 was attributed.

### 2.5. Experimental Conditions

In a single session, patients were tested for PI and RF first in OFF medication (OFF) and then in pharmacological ON medication (ON). In each pharmacological condition, patients were tested in four STN-DBS conditions: DBS of the more affected side (DBS more), DBS of the less affected side (DBS less), bilateral DBS (DBS bilateral), and off DBS (DBS off). More and less affected sides of DBS were defined, respectively, as DBS contralateral and ipsilateral to limbs showing higher scores in the UPDRS part III. The ON-medication recordings were performed one hour after the intake of each subject’s usual dose of levodopa. DBS conditions were set apart by 20 min, and their order was randomized for each participant to exclude a possible bias due to motor learning.

### 2.6. Statistical Analysis

PI, RF, and UPDRS values were tested for normality using the Shapiro–Wilk test on the studentized residuals. Three two-way repeated-measures ANOVAs (rmANOVAs) with “medication” (ON, OFF), and “stimulation” (DBS more, DBS less, DBS bilateral, DBS off) were performed on UPDRS-III, PI, and RF scores. Post hoc analyses were carried out with paired T-tests. Type I errors in testing of multiple pairwise comparisons were controlled for with the false-discovery rate (FDR) correction (maximum acceptable FDR 0.05). *p* values < 0.05 were considered significant. Sphericity in data distribution was verified via Mauchly’s tests, and the Greenhouse–Geisser correction was applied when necessary. Statistical analysis was performed with SPSS Version 25.0.

## 3. Results

Data are expressed as mean ± standard deviation unless otherwise specified. Table 1 shows the patients’ group characteristics.

### 3.1. UPDRS Part III

We found a non-significant “stimulation × medication” interaction on UPDRS part-III (F_3,5_ = 1.33, *p* = 0.3) but a significant effect of “medication” (F_1,5_ = 30.74, *p* < 0.005) and stimulation (F_3,15_ = 22.54, *p* < 0.01).

#### 3.1.1. Bilateral DBS

UPDRS part-III was significantly higher in OFF (51.5 ± 6.6) compared to ON medication (37.1 ± 6.1, t = 5.5, *p* < 0.01, q < 0.01). Also, UPDRS part-III was significantly higher in the DBS-off condition (56.8 ± 20.4) compared with DBS bilateral (32.8 ± 11.6, t = 5.8, *p* < 0.01).

#### 3.1.2. Unilateral DBS

UPDRS part-III was not significantly different when comparing DBS more and DBS less (43.7 ± 16.2 vs. 44.7 ± 13.1, t = 0.7, *p* = 0.5). Furthermore, UPDRS part-III was significantly lower in both DBS more (t = 5.9, *p* = 0.01) and DBS less (t = 3.7, *p* = 0.02) compared with the DBS-off condition. Finally, the UPDRS-III score was significantly higher in both DBS more (t = 3.3, *p* = 0.03) and DBS less (t = 4.7, *p* < 0.01) compared with DBS bilateral (Figure 1).

### 3.2. Postural Instability

Numeric results are reported in Table 2. We found a significant “stimulation × medication” interaction on PI.

#### 3.2.1. Bilateral DBS

In OFF medication, PI values were not significantly different between conditions. In ON medication, PI was significantly higher in DBS off (9.3 ± 4.3) compared to DBS bilateral (4.5 ± 2.0). When testing for the effect of medication, we found that PI was not significantly different between OFF and ON medication in DBS bilateral (4.8 ± 2.2 vs. 4.5 ± 2.0), whereas PI was significantly lower in OFF compared to ON medication in DBS-off (5.0 ± 2.6 vs. 9.3 ± 4.3) conditions.

#### 3.2.2. Unilateral DBS

In ON medication, PI was significantly higher in DBS more (6.8 ± 2.9) compared with DBS less (5.9 ± 2.6). Also, for ON medication, PI was significantly lower in both DBS more and DBS less compared with the DBS-off condition, and there was a trend for higher values in both DBS more and DBS less than DBS bilateral. Finally, for ON medication, PI was not significantly different between OFF and ON medication in the DBS-more condition (6.4 ± 2.7 vs. 6.8 ± 2.9), whereas a trend was observed for lower values in OFF than ON in the DBS-less (4.1 ± 1.7 vs. 5.9 ± 2.6) condition (Table 2) (Figure 2).

### 3.3. Risk of Falls

Numeric results are reported in Table 2. We found a significant “stimulation × medication” interaction on RF.

#### 3.3.1. Bilateral DBS

For OFF medication, RF values were significantly lower in DBS off (4.2 ± 1.9) compared with DBS bilateral (5.6 ± 2.4). For ON medication, RF was significantly higher in DBS off (10.1 ± 4.5) compared with DBS bilateral (5.1 ± 2.3). When testing for the effect of medication, we found that RF was not significantly different between OFF and ON medication in DBS-bilateral conditions (5.6 ± 2.4 vs. 5.0 ± 2.3), whereas RF was significantly higher in ON compared to OFF medication in the DBS-off condition (10.1 ± 4.5 vs. 4.2 ± 1.9).

#### 3.3.2. Unilateral DBS

For OFF medication, RF was significantly higher in DBS more (10.3 ± 4.5) compared to DBS less (6.7 ± 3.4). Also, for OFF medication, RF values were significantly higher in DBS more compared with the DBS-off condition, whereas a trend was only found for higher values in DBS less compared to DBS off. For ON medication, RF was significantly higher in DBS more (8.9 ± 4.2) than DBS less (6.9 ± 3.2) and DBS bilateral, and RF was higher in DBS less compared to DBS bilateral, but this difference did not reach statistical significance. Also, for ON medication, RF was significantly lower in both DBS more and DBS less compared with DBS off. We found that RF was not significantly different between OFF and ON medication in the DBS-less condition (6.7 ± 3.4 vs. 6.9 ± 3.2), whereas RF was significantly higher in OFF compared to ON medication in the DBS-more condition (10.3 ± 4.5 vs. 8.9 ± 4.2) (Table 2) (Figure 3).

### 3.4. Summary of Results

UPDRS part-III values were significantly lower in ON medication compared to OFF medication. Also, UPDRS part-III values were significantly lower in DBS bilateral compared to all other DBS conditions and were significantly higher in DBS off compared to all other DBS conditions.

The most favorable (i.e., lowest) PI and RF scores were noted in patients who were OFF medication and in the DBS-off condition. Within the DBS-off state, patients exhibited worse (i.e., increased) PI and RF values when they were ON medication compared to OFF medication.

#### 3.4.1. Bilateral DBS

When considering OFF medication, while DBS did not notably affect PI, the DBS-bilateral condition was the least detrimental among DBS conditions for RF. Conversely, in the context of ON medication, the DBS-bilateral condition significantly enhanced both PI and RF.

#### 3.4.2. Unilateral DBS

When considering OFF medication, the DBS-more setting registered the worst (i.e., highest) RF, while DBS less exhibited intermediate outcomes. On the other hand, during ON medication, both PI and RF were improved (i.e., reduced) by both unilateral DBS settings. The DBS-less condition showed significantly better outcomes than DBS more but tended to have results that were not as optimal as the bilateral-DBS condition.

The effects of DBS, medication, and their interaction on PI and RF values were independent from motor improvement, as assessed by UPDRS part-III.

## 4. Discussion

In the present study, we investigated the effects of the interaction between pharmacological dopaminergic treatment and STN-DBS on balance control in advanced PD. We found that PD patients had the best motor performance, as tested via UPDRS part-III for ON medication and DBS bilateral. When in DBS off, dopaminergic treatment worsened both PI and RF, and when in OFF medication, DBS did not change PI but worsened RF. For ON medication, PI and RF significantly improved after turning the DBS on, regardless of the stimulation condition. However, no combination between pharmacological treatment and DBS was able to induce a significant improvement in postural control with respect to the OFF-medication/DBS-off conditions.

This is the first study that systematically assessed PI and RF in all medication and DBS conditions in PD in a single session. Previous studies reported conflicting results on the relationship between dopaminergic treatment and motor balance, showing either an improvement [23,24], no effect [25], or even a worsening [10,26]. Our observation that dopaminergic medication improves motor performance but worsens PI and RF may be explained by a reduction in axial rigidity without an associated improvement in postural control [27]. This result also suggests that balance control in advanced PD cannot be due to a dopaminergic deficit only, thus confirming previous observations [25,26,27,28,29,30].

Although several studies have investigated the effect of STN-DBS on balance control, producing conflicting results, few used objective posturometric measurements [6,12,13]. While studies that compared DBS effects before and shortly after surgery may be biased by the surgical procedure itself [31], studies that assessed the effect of DBS in a single session after surgery found an improvement in postural control, which is consistent with our results [24,26]. However, these studies were limited by evaluation performed only in bilateral DBS [26] or in OFF-medication conditions [24]. To the best of our knowledge, only one study investigated the effect of DBS on postural control while ON medication using the Biodex Stability System (BSS) and found a non-significant improvement in postural sway in bilateral DBS compared to DBS off [11]. However, patients were tested with eyes closed, and since an abnormal visual control of balance is thought to contribute to postural instability in PD, performing the tests with the eyes closed may have limited the BSS sensitivity to probe postural control in this population [32].

In line with previous studies, our results showed that, whereas STN-DBS improved PI and RF when ON medication, dopaminergic medication alone had a negative impact on both indices [12,33]. A DBS-associated reduction in PI and RF when ON medication may, in part, account for balancing out the negative effect of dopaminergic medication on postural control. These observations hint that the effect of STN-DBS on postural control may recognize non-dopaminergic mechanisms. Supporting this hypothesis, several lines of evidence suggested that the subthalamo-nigro-pedunculopontine pathway plays a central role in determining balance control in humans, and STN-DBS might improve axial symptoms by modulating non-dopaminergic descending pathways directed at the pedunculopontine nucleus [12,34]. The observation that DBS bilateral showed better PI and RF control than both unilateral DBS settings suggests that a balanced activity between the two STNs is important for postural control mechanisms in both static and dynamic conditions. However, how do we conciliate this hypothesis with the finding that less-affected-side DBS produced better PI and RF control than more-affected-side DBS? If the effect of STN-DBS on PI and RF is mediated only by the silencing of a hyperactive STN, the more-affected-side DBS should produce better balance than DBS of the less affected side. Instead, when OFF medication (i.e., when abnormal STN activity should be maximal), more-affected-side DBS proved to worsen RF compared to DBS-off and other DBS conditions including the less-affected side. The question remains unanswered and warrants further investigation with larger study samples.

The main limitation of our study is represented by the small sample size; this was due to the difficulty in recruiting patients with appropriate features, such as long disease duration, high disability, and DBS implantation. Therefore, although the effects we found were consistent across patients and were statistically significant, we acknowledge that our findings are preliminary and need confirmation in future larger-scale studies. Since we used a cross-sectional design, we cannot draw any conclusions about the long-term effect of dopaminergic therapy and DBS on PI and RF. However, the cross-sectional design excluded bias produced by disease progression and day-to-day variability in symptoms and allowed us to compare different medication and DBS setups, providing useful information to guide the therapeutic management of PD patients.

In conclusion, improvements in motor performance produced by dopaminergic medication alone and DBS alone come at the cost of reduced postural control. However, bilateral DBS seems to balance out the negative effect of dopaminergic medication on postural control. Bilateral DBS, when ON medication, proved to be the best option to obtain a maximal improvement in motor performance at a minimal cost in terms of balance control. Finally, the consistency of our results between subjects suggests that the BSS, in addition to predicting falls in patients implanted with DBS [35], may be a promising tool to assess the medication/stimulation combination that produces the best postural control in PD.

## Figures and Tables

**Figure 1 brainsci-13-01243-f001:**
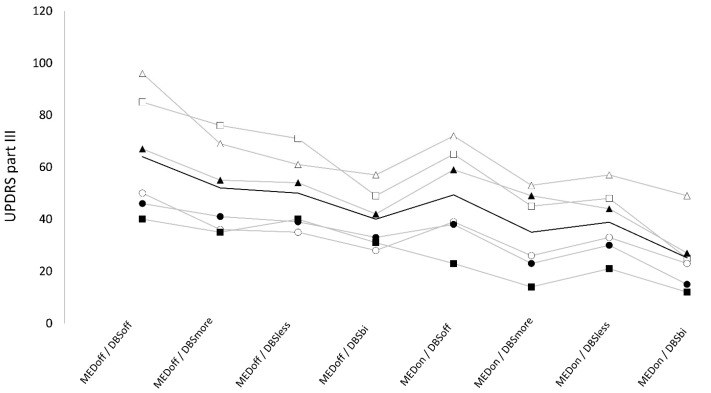
UPDRS-III values across conditions. Data points and grey lines represent single subject mean values. Black line represents mean values across subjects. Conditions: OFF medication (MEDoff), ON medication (MEDon), off DBS (DBSoff), bilateral subthalamic nucleus (STN) deep brain stimulation (DBS) (DBSbi), unilateral STN DBS of the more affected side (DBSmore), and of the less affected side (DBSless).

**Figure 2 brainsci-13-01243-f002:**
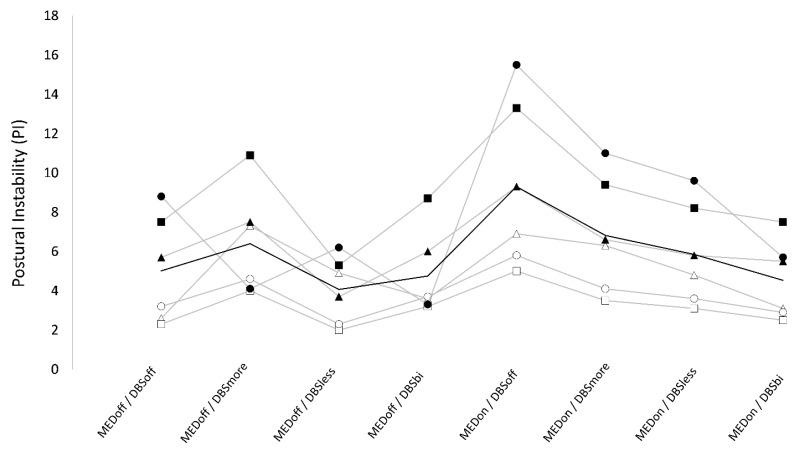
PI values across conditions. Data points and grey lines represent single subject values. Black line represents mean values across subjects. Conditions: OFF medication (MEDoff), ON medication (MEDon), off DBS (DBSoff), bilateral subthalamic nucleus (STN) deep brain stimulation (DBS) (DBSbi), unilateral STN DBS of the more affected side (DBSmore), and of the less affected side (DBSless).

**Figure 3 brainsci-13-01243-f003:**
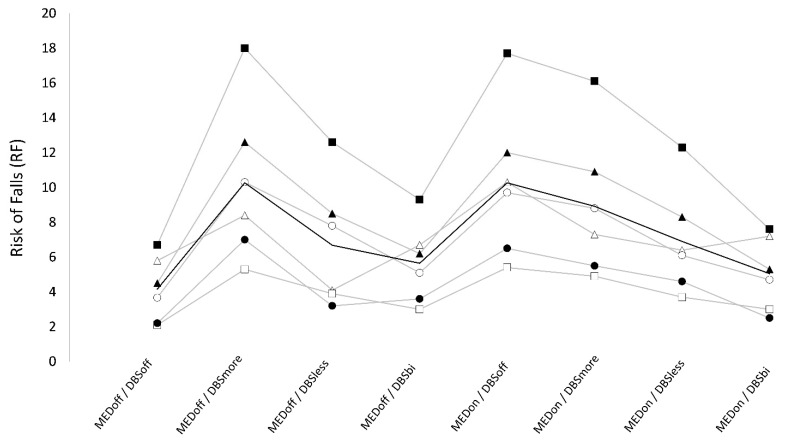
RF values across conditions. Data points and grey lines represent single subject values. Black line represents mean values across subjects. Conditions: OFF medication (MEDoff), ON medication (MEDon), off DBS (DBSoff), bilateral subthalamic nucleus (STN) deep brain stimulation (DBS) (DBSbi), unilateral STN DBS of the more affected side (DBSmore), and of the less affected side (DBSless).

**Table 1 brainsci-13-01243-t001:** Patients’ demographics and clinical information.

Subjects	Age (ys)	Gender	Diseases Duration (ys)	UPDRS III OFF	UPDRS III ON	More Affected Side	LEDDS
01	72	F	20	96	72	R	240
02	65	F	14	85	65	L	250
03	69	M	19	50	39	R	436
04	58	F	18	40	23	R	620
05	62	M	12	46	38	L	540
06	72	M	20	67	59	R	500
Avg	66 ± 6	F/M 3/3	17 ± 3	64 ± 23	49 ± 19	R/L 4/2	416 ± 147

**Table 2 brainsci-13-01243-t002:** Postural instability and risk of falls main results. DV: dependent variable; df: degree of freedom; F: F-test; T: *t*-test; *p*: *p* value; adj. *p*: adjusted *p* values with false discovery rate (q values).

2-Way ANOVA						
**Factor**	**DV**		**df**	**F**	** *p* **	
**Postural Instability (PI)**						
Stim	PI		3;15	5.51	0.009	
Med	PI		1;5	5.4	0.068	
Stim * Med	PI		3;15	12.48	<0.001	
**Risk of falls (RF)**						
Stim	RF		3;15	12.92	0.01	
Med	RF		1;5	18.61	0.008	
Stim × Med	RF		3;15	22.7	0.003	
**Pairwise comparisons**						
	**PI**			**RF**		
	** *t* **	** *p* **	**adj. *p***	** *t* **	** *p* **	**adj. *p***
MEDoff/DBSoff vs. MEDoff/DBSmore	1.04	0.345	0.467	4.62	0.006	0.014
MEDoff/DBSoff vs. MEDoff/DBSless	1.34	0.239	0.353	2.41	0.061	0.074
MEDoff/DBSoff vs. MEDoff/DBSbil.	0.25	0.810	0.810	5.08	0.004	0.014
MEDoff/DBSoff vs. MEDon/DBSoff	5.99	0.002	0.035	4.98	0.004	0.014
MEDoff/DBSoff vs. MEDon/DBSmore	4.04	0.010	0.035	3.98	0.011	0.018
MEDoff/DBSoff vs. MEDon/DBSless	2.94	0.032	0.075	3.68	0.014	0.023
MEDoff/DBSoff vs. MEDon/DBSbil.	0.90	0.411	0.523	4.16	0.009	0.018
MEDoff/DBSmore vs. MEDoff/DBSless	2.24	0.075	0.132	6.84	0.001	0.014
MEDoff/DBSmore vs. MEDoff/DBSbil.	3.54	0.017	0.052	4.46	0.007	0.014
MEDoff/DBSmore vs. MEDon/DBSoff	1.66	0.157	0.244	0.33	0.758	0.758
MEDoff/DBSmore vs. MEDon/DBSmore	0.32	0.763	0.791	9.59	0.000	0.006
MEDoff/DBSmore vs. MEDon/DBSless	0.44	0.678	0.756	5.77	0.002	0.014
MEDoff/DBSmore vs. MEDon/DBSbil.	2.29	0.071	0.132	4.07	0.010	0.018
MEDoff/DBSless vs. MEDoff/DBSbil.	0.71	0.508	0.592	1.35	0.235	0.253
MEDoff/DBSless vs. MEDon/DBSoff	4.38	0.007	0.035	4.46	0.007	0.014
MEDoff/DBSless vs. MEDon/DBSmore	4.69	0.005	0.035	5.11	0.004	0.014
MEDoff/DBSless vs. MEDon/DBSless	3.41	0.019	0.053	0.48	0.654	0.678
MEDoff/DBSless vs. MEDon/DBSbil.	0.78	0.473	0.576	1.54	0.184	0.207
MEDoff/DBSbil. vs. MEDon/DBSoff	2.87	0.035	0.075	4.93	0.004	0.014
MEDoff/DBSbil. vs. MEDon/DBSmore	1.75	0.141	0.233	3.61	0.015	0.024
MEDoff/DBSbil. vs. MEDon/DBSless	1.03	0.350	0.467	2.77	0.039	0.053
MEDoff/DBSbil. vs. MEDon/DBSbil.	0.41	0.702	0.756	1.91	0.115	0.134
MEDon/DBSoff vs. MEDon/DBSmore	4.06	0.010	0.035	3.29	0.022	0.030
MEDon/DBSoff vs. MEDon/DBSless	4.96	0.004	0.035	5.68	0.002	0.014
MEDon/DBSoff vs. MEDon/DBSbil.	4.30	0.008	0.035	4.56	0.006	0.014
MEDon/DBSmore vs. MEDon/DBSless	5.06	0.004	0.035	4.45	0.007	0.014
MEDon/DBSmore vs. MEDon/DBSbil.	3.31	0.021	0.054	3.33	0.021	0.030
MEDon/DBSless vs. MEDon/DBSbil.	2.39	0.063	0.125	2.56	0.051	0.065

## Data Availability

Data is available upon request.

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
