# Peer review of "Postural Instability and Risk of Falls in Patients with Parkinson’s Disease Treated with Deep Brain Stimulation: A Stabilometric Platform Study"

_brainsci, 2023, doi:10.3390/brainsci13091243_

Round 1
Reviewer 1 Report
The authors examined the effects of DBS and medication (on and off) on postural instability and fall risk in 6 patients with PD who had received DBS at least 1 year prior. Postural instability has not been well managed by either medication or DBS so this was an attempt to separate out possible interaction effects.
a few concerns - one is the use of the DBS more and DBS less measures. There is little information about this approach and results have been mixed (e.g., Lin et al. 2021 npj/Parkinson's disease). It might be better to separate these out as more exploratory analyses from the overall look at DBS and medication on/off combinations. It is very difficult to follow the results with so many comparisons.
The authors indicate that 6 patients is quite small. It does limit the ability to extend these findings. Especially with all of the treatment combinations that are measured.
The abstract is confusing in how results are presented. Suggest shorter sentences so results can be comprehended in more bite-size pieces.
page 9, line 232 - you refer to BSS - what is this?
Some reference(s) in the background section to describe testing under the DBS more and less conditions and for what outcomes would be informative. What has this strategy been used to test before.
Author Response
Response to Reviewer 1 Comments
Point 1: a few concerns - one is the use of the DBS more and DBS less measures. There is little information about this approach and results have been mixed (e.g., Lin et al. 2021 npj/Parkinson's disease). It might be better to separate these out as more exploratory analyses from the overall look at DBS and medication on/off combinations. It is very difficult to follow the results with so many comparisons.
Response 1: We understand the reviewer concerns and have changed our manuscript accordingly. In the revised manuscript, we have restructured the entire Results section in paragraph to provide a clearer distinction between the outcomes associated with bilateral and unilateral DBS.
Point 2: The authors indicate that 6 patients is quite small. It does limit the ability to extend these findings. Especially with all of the treatment combinations that are measured.
Response 2: We agree that the sample size is modest, which may limit the generalizability of our findings as stated in the imitation paragraph (page 10, lines 277-281). Nevertheless, considering the uniqueness of our cohort, we believe our findings contribute valuable preliminary insights into this area. We have further emphasized the preliminary nature of our findings in the discussion section and called for larger-scale studies to validate and extend our observations (page 10, lines 279-281).
Point 3: The abstract is confusing in how results are presented. Suggest shorter sentences so results can be comprehended in more bite-size pieces.
Response 3: Thank you for your valuable feedback. We have rephrased the abstract for clarity. Specifically, we have shortened the sentences to ensure that the results are presented in a more comprehensible manner. We believe that the revised abstract now communicates our findings more effectively and concisely.
Point 4: page 9, line 232 - you refer to BSS - what is this?
Response 4: The “BSS” abbreviation stands for “Biodex Stability System (BSS)” (see page 2, line 84). We have now expanded "BSS" upon its first mention in the discussion.
Point 5: Some reference(s) in the background section to describe testing under the DBS more and less conditions and for what outcomes would be informative. What has this strategy been used to test before.
Response 5: We acknowledge the importance of providing context for testing the effect of unilateral DBS. We have supplemented the background section with relevant references on this topic from previous studies (page 2, lines 50-56)
Reviewer 2 Report
The article titled "Postural Instability and Risk of Falls in Patients with Parkinson’s Disease Treated with Deep Brain Stimulation: A Stabilometric Platform Study" explores the complex interplay between medication and deep brain stimulation (DBS) in Parkinson’s disease (PD) patients' postural control and fall risk. The authors investigate this relationship using a comprehensive approach that involves evaluating postural instability (PI) and risk of falls (RF) across various medication and DBS conditions. While the study offers valuable insights into the balance control mechanisms in PD, several aspects of the article warrant critical examination.
The introduction of the article sets the stage by outlining the significance of postural control in PD, emphasizing the multifactorial origins of PI and RF. This introduction successfully establishes the context for the study's focus, but it could benefit from more in-depth discussions on the existing literature, particularly the variability in outcomes of previous research on the effects of DBS and medication on postural control.
The methodology section provides a detailed overview of participant selection, assessments, and statistical analyses. However, the small sample size (six patients) is a major limitation that affects the study's generalizability and statistical power. This limitation is acknowledged by the authors, but it is essential to emphasize that the findings should be interpreted with caution due to the restricted sample size.
While the study's systematic approach is commendable, the results section could be presented more concisely. The presentation of results is exhaustive, with tabular data and multiple comparisons. A more streamlined presentation would enhance readability and help the reader focus on the main findings. The data indicate a nuanced interaction between medication and DBS on PI and RF, but the extent of the clinical implications for PD patients remains somewhat unclear.
The discussion section provides insights into the study's findings, linking them to the broader context of PD research. The authors suggest that the balance of dopaminergic therapy and DBS is crucial for optimal postural control. However, the discussion could delve further into the potential neurobiological mechanisms underlying the observed effects, such as the involvement of the subthalamo-nigro-pedunculopontine pathway.
In conclusion, the study offers valuable insights into the intricate relationship between medication and DBS on postural control in PD patients. However, the limitations of the small sample size and the need for further exploration of the underlying mechanisms should be acknowledged. Future research with larger sample sizes and a more comprehensive evaluation of neurobiological factors could provide a more robust understanding of the observed effects. The article's systematic approach and methodological rigor contribute to the field of PD research, but caution is warranted in interpreting the findings due to the sample size limitations.
Subject-Verb Agreement: In the sentence "DBS associated reduction of PI and RF when ON-medication may in part account for balancing out the negative effect of dopaminergic medication on postural control," the subject "reduction" should agree with the verb "accounts." It should be "DBS-associated reduction ... accounts."
Punctuation: In the sentence "Therefore, albeit the effects were consistent across patients and were statistically significant, they need confirmation in future work," the comma before "albeit" is not needed. It should read "Therefore, although the effects were consistent..."
Word Choice: The phrase "produce better PI and RF than monolateral DBS" should be "produce better PI and RF control than monolateral DBS" for clarity.
Inconsistent Capitalization: The term "bilateral-DBS" should have consistent capitalization throughout the text. It's sometimes written as "bilateral-DBS" and at other times as "bilateral DBS."
Misplaced Modifier: In the sentence "When OFF-medication, DBS did not affect postural instability, while all DBS-on conditions (i.e., bilateral, more and less affected side) worsened RF compared to DBS-off with the highest RF in DBS-more and lowest in bilateral-DBS," the modifier "with the highest RF in DBS-more and lowest in bilateral-DBS" seems to be modifying "DBS-off," but it is meant to modify "DBS-on conditions." Consider rephrasing for clarity.
Author Response
Response to Reviewer 2 Comments
Point 1: The introduction of the article sets the stage by outlining the significance of postural control in PD, emphasizing the multifactorial origins of PI and RF. This introduction successfully establishes the context for the study's focus, but it could benefit from more in-depth discussions on the existing literature, particularly the variability in outcomes of previous research on the effects of DBS and medication on postural control.
Response 1: We agree that a more in-depth discussion on the existing literature, especially focusing on the variability of outcomes in prior research, would strengthen the paper's introduction. We have enriched the introduction by discussing the variability in outcomes of previous research on the effects of DBS and medication on postural control. We have also emphasized how different methodologies might contribute to such inconsistencies (page 2, lines 44-58).
Point 2: The methodology section provides a detailed overview of participant selection, assessments, and statistical analyses. However, the small sample size (six patients) is a major limitation that affects the study's generalizability and statistical power. This limitation is acknowledged by the authors, but it is essential to emphasize that the findings should be interpreted with caution due to the restricted sample size.
Response 2: We agree that the sample size is modest, which may limit the generalizability of our findings as stated in the imitation paragraph (page 10, lines 277-281). We have further emphasized the preliminary nature of our findings in the discussion section and called for larger-scale studies to validate and extend our observations (page 10, lines 279-281).
Point 3: While the study's systematic approach is commendable, the results section could be presented more concisely. The presentation of results is exhaustive, with tabular data and multiple comparisons. A more streamlined presentation would enhance readability and help the reader focus on the main findings. The data indicate a nuanced interaction between medication and DBS on PI and RF, but the extent of the clinical implications for PD patients remains somewhat unclear.
Response 3: We appreciate the reviewer’s feedback regarding the presentation of results. We have condensed the presentation of the entire Results section and separated the results regarding the effects of bilateral DBS from the results of unilateral DBS.
Point 4: The discussion section provides insights into the study's findings, linking them to the broader context of PD research. The authors suggest that the balance of dopaminergic therapy and DBS is crucial for optimal postural control. However, the discussion could delve further into the potential neurobiological mechanisms underlying the observed effects, such as the involvement of the subthalamo-nigro-pedunculopontine pathway.
Response 4: We appreciate the reviewer's emphasis on discussing neurobiological mechanisms. Our manuscript already addresses the potential mechanisms underlying the observed effects, including the role of the subthalamo-nigro-pedunculopontine pathway (page 10, lines 257-276). We believe that further speculation might go beyond our study's scope.
Point 5: In conclusion, the study offers valuable insights into the intricate relationship between medication and DBS on postural control in PD patients. However, the limitations of the small sample size and the need for further exploration of the underlying mechanisms should be acknowledged. Future research with larger sample sizes and a more comprehensive evaluation of neurobiological factors could provide a more robust understanding of the observed effects. The article's systematic approach and methodological rigor contribute to the field of PD research, but caution is warranted in interpreting the findings due to the sample size limitations.
Response 5: We appreciate the reviewer’s constructive feedback. As discussed in the previous comment, we acknowledge the limitation of the small sample size. In the revised discussion, we have further emphasized that our findings should be interpreted with caution due to the limited sample size (page 10, lines 279-281).
Point 6: Subject-Verb Agreement: In the sentence "DBS associated reduction of PI and RF when ON-medication may in part account for balancing out the negative effect of dopaminergic medication on postural control," the subject "reduction" should agree with the verb "accounts." It should be "DBS-associated reduction ... accounts."
Response 6: We would like to thank the reviewer for pointing out the subject-verb agreement in the sentence highlighted. However, upon further review, when "may" precedes another verb, the following verb should be in its base form. Thus, "may account" is the grammatically correct form.
Point 7: Punctuation: In the sentence "Therefore, albeit the effects were consistent across patients and were statistically significant, they need confirmation in future work," the comma before "albeit" is not needed. It should read "Therefore, although the effects were consistent..."
Response 7: We agree with the reviewer. The punctuation error has been rectified (page 19, line 279).
Point 8: Word Choice: The phrase "produce better PI and RF than monolateral DBS" should be "produce better PI and RF control than monolateral DBS" for clarity.
Response 8: We agree with the reviewer, and we have modified the text accordingly (page 9, line 267 and page 10, line 270).
Point 9: Inconsistent Capitalization: The term "bilateral-DBS" should have consistent capitalization throughout the text. It's sometimes written as "bilateral-DBS" and at other times as "bilateral DBS."
Response 9: The terms “bilateral DBS” and "DBS-bilateral" have been reviewed for consistent capitalization throughout the manuscript.
Point 10: Misplaced Modifier: In the sentence "When OFF-medication, DBS did not affect postural instability, while all DBS-on conditions (i.e., bilateral, more and less affected side) worsened RF compared to DBS-off with the highest RF in DBS-more and lowest in bilateral-DBS," the modifier "with the highest RF in DBS-more and lowest in bilateral-DBS" seems to be modifying "DBS-off," but it is meant to modify "DBS-on conditions." Consider rephrasing for clarity.
Response 10: The results section, including the mentioned sentence, has been entirely rephrased for clarity (see Point 1 from Rev #1, and Point 3 from Rev #2).
Round 2
Reviewer 1 Report
authors have been responsive to critiques, no other concerns
Reviewer 2 Report
The authors have paid attention to all my comments. Thanks so much.